# Electroretinography as a Biomarker to Monitor the Progression of Stargardt Disease

**DOI:** 10.3390/ijms232416161

**Published:** 2022-12-18

**Authors:** Jana Sajovic, Andrej Meglič, Marko Hawlina, Ana Fakin

**Affiliations:** Eye Hospital, University Medical Centre Ljubljana, Grablovičeva 46, 1000 Ljubljana, Slovenia

**Keywords:** Stargardt disease, *ABCA4* gene, electroretinographic biomarkers, electroretinography, electrophysiology, retinal dystrophy, S-cone ERG, DA 0.01 ERG b-wave, DA 3.0 ERG a-wave

## Abstract

The aim of the present study is to determine how electroretinographic (ERG) responses reflect age-related disease progression in the Stargardt disease (STGD1). The prospective comparative cohort study included 8 patients harboring two null *ABCA4* variants (Group 1) and 34 patients with other ABCA4 genotypes (Group 2). Age at exam, age at onset, visual acuity (VA) and ERG responses were evaluated. The correlation between ERG responses and age in each patient group was determined using linear regression. A Mann-Whitney U Test was used to compare the median values between the groups. Age of onset was significantly earlier in Group 1 than in Group 2 (8 vs. 18), while disease duration was similar (13 vs. 12 years, i.e., advanced stage). Group 1 had significantly worse VA and lower ERG responses. ERG responses that significantly correlated with age in Group 1 were DA 0.01 and 3.0 ERG, which represented a retinal rod system response. The only ERG response that significantly correlated with age in Group 2 was the S-cone ERG. The observed difference was likely due to early cone loss occurring in double-null patients and slower photoreceptor loss in patients with other genotypes. The results suggest that specific ERG responses may be used to detect double-null patients at an early stage and monitor STGD1 disease progression in patients with specific genotypes.

## 1. Introduction

Stargardt disease (STGD1), also known as fundus flavimaculatus or ABCA4 retinopathy, is a progressive disorder of the retinal pigment epithelium (RPE) and photoreceptors caused by bi-allelic variants of the *ABCA4* gene [1,2,3,4,5,6]. It ranges from macular dystrophy, which only affects central vision (mild phenotype), to generalized photoreceptor dystrophy, which can lead to blindness (severe phenotype) [7]. Onset is most common in childhood or adolescence and least frequent in later adulthood, with a worse prognosis usually associated with an earlier initiation of disease symptoms [1,2]. Cone degeneration is more severe and occurs before the degeneration of rods, which corresponds to the clinical signs and the course of the disease [8,9]. The ABCA4 protein is mainly expressed in the photoreceptor outer segments. In rods, it is localized on the rims and incisures of enclosed disk membranes, while in cones, it is present on the lamellar margins and is in contact with extracellular space [4,10,11,12]. Additionally, the protein is found in the internal membranes of the RPE [5]. It functions as a transmembrane transporter of molecules involved in visual transduction, making it an essential part of the visual cycle for removal of toxic vitamin A products [9].

Although the disease is caused by a single gene, the genetic spectrum of patients is extremely heterogenous, as there are >2280 known disease-associated variants in the *ABCA4* gene (www.lovd.nl/ABCA4, accessed on 2 October 2022), which can pair with each other in millions of different combinations [7,13].

Genotype–phenotype correlation studies have shown that a large part of clinical variability in STGD1 is related to the type and severity of the genetic variants. In general, a higher residual ABCA4 function results in a milder phenotype [1]. Patients with two null alleles, in whom the ABCA4 protein is completely absent, have the most severe disease, which is characterized by childhood onset progressive cone–rod dystrophy, which normally leads to blindness in the fifth decade [7,14], while others have a range of milder phenotypes [13,15,16].

To determine the outcome for clinical studies, monitor disease progression, measure the response to treatment, and elucidate disease mechanisms, predictable biomarkers have become a target in the research. In STGD1, the retinal structure has been studied by imaging biomarkers [17,18,19,20,21], while sensitive, reliable, valid, and objective biomarkers enabling the early detection and follow-up of changes in retinal function according to genotypes have not yet been addressed.

Electroretinography (ERG) is a method that can objectively quantify retinal function. It has previously been used to categorize STGD1 in three ERG groups: ERG Group I—disease limited to the macula, ERG Group II—cone dystrophy, and ERG Group III—cone–rod dystrophy. Although the classification was performed without genotype correlations, a possible genetic background behind the differences was suspected [22]. The classification helps determine the patient prognosis, as it has been shown that ERG Group I patients are less likely to develop retina-wide disease [23,24]. A small subgroup of Group I patients and most of Group II and III patients were shown to have a progressive disease [23]; however, year-to-year deterioration within an ERG group is not reflected in the traditional classification.

The aim of the present study is to perform a quantitative analysis of individual ERG responses in patients with different genotypes in order to determine those most suitable for the following of disease progression.

## 2. Results

The clinical characteristics of the two groups are presented in Table 1. Patients in Group 1 were relatively younger than patients in Group 2 (median age 21 vs. 39 years, respectively); however, the difference was not significant. Group 1 had a significantly earlier age of onset than Group 2 (median 8 and 18 years, respectively), whereas disease duration was similar between the groups (median 13 and 12 years, respectively). Group 1 patients had significantly worse visual acuity (VA) than Group 2 (Figure 1).

Although a formal imaging analysis was not part of this study, we observed features of a more severe disease in Group 1 patients, including widespread fundus autofluorescence changes and extensive photoreceptor loss on spectral-domain optical coherence tomography imaging.

### Electrophysiology

Group 1 patients had significantly lower values of all ERG responses in comparison to Group 2 patients. A large field pattern ERG (PERG) P50 amplitude was not detectable in any of the Group 1 patients, while it was detectable in 82% of the Group 2 patients. Moreover, in 24% of the Group 2 patients, the PERG P50 amplitude was within normal values (Table 2). All analyzed scotopic and photopic ERG responses were significantly lower in Group 1 compared to Group 2. The differences between the two groups are represented graphically in Figure 2. According to the traditional ERG classification conducted by Lois et al. [22], all Group 1 patients belonged to ERG Group III, while almost half of the Group 2 patients were in ERG Group I (Table 2).

A simple linear regression analysis of the correlation between age and ERG amplitudes showed different results for the two genotypic groups (Table 3). In Group 1, age significantly correlated with the dark-adapted (DA) 0.01 ERG b-wave (Figure 3C) and DA 3.0 ERG a-wave (Figure 3E), while in Group 2, age significantly correlated with the S-cone ERG amplitude (Figure 4F). Other ERG parameters did not show significant correlations with age in either group.

In Group 1, DA 0.01 ERG b-wave responses were reduced to undetectable after the age of 40 years, whereas DA 3.0 ERG a-wave responses were still detectable above noise level until the age of 50 years (Figure 3C,E, example in Figure 5A,B). In Group 2 patients, only S-cone ERG amplitudes declined significantly with time (Figure 5C,D), whereas other parameters showed little time dependency (Figure 3 and Figure 4).

## 3. Discussion

To our knowledge, this is the first time that most relevant ERG biomarkers were determined for different genetic subgroups of ABCA4 retinopathy, specifically for the double-null genotype and separately for patients with other non-null genotypes. Our results suggest that different ERG responses may be best suited to monitor the progression of STGD1 disease in different patients. This reflects and accounts for the difference in the severity of photoreceptor damage between patients with different ABCA4 genotypes. In each of the two patient subgroups, we were able to isolate the biomarkers, which mostly reflected the dynamic state of the disease. It is very likely that, by dividing the groups based on onset and/or phenotype, the results would be similar. Since both the disease onset as well as phenotype depend on the genotype, we decided to use the division by genotype.

The measurements of PERG and full-field ERG (ffERG) take approximately 1 h, which includes 20 min of dark and 10 min of light adaptation. The addition of the S-cone ERG measurement, which is not included in the standard International Society of Clinical Electrophysiology of Vision (ISCEV) protocol, prolongs the entire exam for a few minutes through the series of tests following light adaptation. In comparison with fundus autofluorescence and spectral-domain optical coherence tomography imaging, which takes less than 5 min, ERG measurements are quite time-consuming and more demanding for a patient. However, they are the most accurate and objective measures of photoreceptor function, and they are thus very valuable in clinical and research settings.

The short-term changes are difficult to detect if the measured parameter is either too reduced or too preserved (floor and ceiling effect, respectively). The ERG biomarkers that best reflected disease progression in the double-null-patient group were DA 0.01 and 3.0 ERG, which represent rod system functions. This genotype is associated with an absent ABCA4 protein and results in early onset cone-rod dystrophy with an early and severe loss of cone photoreceptor function [7,14,25]. In this situation, cone responses are difficult to quantify, but rod responses are still measurable and suitable for monitoring purposes. For other genotypes associated with possible residual ABCA4 function leading to a milder phenotype, the best ERG biomarker appears to be the S-cone ERG. Patients with milder phenotypes often exhibit good peripheral photoreceptor function until the late stages [13,22,24]. Retinal function may be so stable that there is no measurable loss of most ERG parameters during a short-term follow-up treatment. However, photoreceptor loss does occur over time in the majority of patients [23]. Interestingly, S-cone ERG was more sensitive than traditional LA ERG parameters reflecting L/M cone loss. This is an important result that should be considered in clinical practices and trials, where the S-cone may be considered as an additional biomarker for patients with relatively preserved ABCA4 functions.

The present study may provide some insights into the disease’s pathogenesis, especially in relation to cone photoreceptors. It has been suggested that mechanisms involved in ABCA4 retinopathy include secondary photoreceptor loss due to RPE damage, as well as direct cone toxicity. The reason for primary cone loss is still not entirely understood in the field of study [8]; however, it is thought to be linked to the anatomical specifics of cone photoreceptors that allow the accumulation of toxic bisretinoids on their outer membranes [8,26].

According to the photopigment absorption spectra, cones can be divided into long-wave-sensitive (L), middle-wave-sensitive (M), and short-wavelength-sensitive (S) cones [27]. The morphology of S-cones is distinct from that of L- and M-cones, and resembles that of rods. They are smaller and have shorter outer segments, while the inner segments are long, which results in a cylindrical rather than conical shape. S-cones are absent in the foveal center and are most abundant at a 100 μm eccentricity [28]. Moreover, L- and M-cones together constitute approximately 90% of the total number of cones, while S-cones only represent approximately 7–10% [28,29]. For this reason, S-cone analysis with ERG is very challenging. In STGD1, cone system function is usually assessed by standard light-adapted (LA) ERG testing, which is dominated by L- and M-cone responses, where S-cone-mediated retinal activity is not directly visible. S-cone ERG is not routinely applied in the clinical evaluation of STGD1 patients [30], and no specific impairment of S-cones that differs from that of L- and M-cones has ever been shown in STGD1 patients to date. It is possible that S-cones are affected in a different fashion than L- and M-cones. It would be interesting to re-evaluate the localization and cell-specific function of the ABCA4 protein in different types of cones.

Even though several different therapeutic strategies have been evaluated in clinical studies [31,32,33,34,35], no therapy form has been registered yet for the treatment of STGD1. Clinical studies require sensitive and objective biomarkers that enable the detection of minor changes in retinal function. Our study substantiated the value of ERG in assessing the retinal function of ABCA4 patients and contributed to individualized genotype prediction. While fundus autofluorescence and optical coherent tomography are traditionally used to assess the structural signs of photoreceptor loss, ERG is the only method that objectively detects retinal dysfunction. It can detect whether photoreceptor dysfunction is confined to the macula or has spread to the whole retina, even before structural signs appear. Moreover, it can delineate the dysfunction of different photoreceptor types, which is not possible for the imaging methods used at present.

### Study Strength and Limitations

The main strength of the study was the use of a quantitative approach to study ERG responses. Previous ERG classification methods only categorize patients into three ERG groups [22], which does not account well for the genotype or progressive nature of the disease. In this study, the parameters were correlated with age and genotype groups, which allowed for the observation of intermediate phenotypes between the classical ERG groups. Another strength was the use of PERG with the larger stimulus field, which can produce recordable responses for most patients. Although PERG applied to the standard stimulus field provides greater sensitivity and specificity of early maculopathy, the larger stimulus field is more suitable for the longer follow-up treatment of STGD1 patients, as previously demonstrated in a group of STGD1 patients, where responses were detectable in 86% of patients using the large field and in only 22% using the standard field [36]. Furthermore, our study encompassed the largest spectrum of ERG parameters, including the S-cone ERG, which was analyzed in the largest group of STGD1 patients to date. The genetic stratification allowed for a systematic analysis that showed unequivocal differences between the two different genetic groups of STGD1 patients.

Nevertheless, the main limitations of our study were the small number of patients and the heterogenous genetic structure of the Group 2 patients. Namely, patients in Group 2 had various combinations of variants, with the exception of two null variants, resulting in a range of retinal phenotypes. However, the analysis of further stratified subgroups was not possible, due to an even smaller number of patients and the loss of statistical power. A larger, possibly multicentric study including more patients with specific variants, in the early stages of the disease and with longitudinal follow-up, is likely to also reveal age-dependence for other ERG parameters, including PERG P50, which describes the photoreceptor function in the macula—the primary target of the disease. Another limitation was also the use of a relatively simple statistical approach. A principal component analysis might reveal the further classification of disease progression.

## 4. Materials and Methods

### 4.1. Patients

The prospective study included 42 STGD1 patients recruited from the Eye Hospital at the University Medical Center in Ljubljana, Slovenia. The inclusion criteria were a clinical phenotype consistent with ABCA4 retinopathy and at least two possible disease-causing variants in *ABCA4* detected via genetic analysis. The patients were divided into two groups based on their genotypes. Group 1 consisted of eight patients harboring null variants (three male, five female) (i.e., “double-null patients”), and Group 2 of 34 patients with all other genotypes (10 male, 24 female). A variant was considered to be null if it was either a stop variant, a frame-shifting variant resulting in a stop codon, or was a splicing and missense variant previously shown to behave as null (p.[(Leu541Pro;Ala1038Val)], p.(Glu1022Lys), p.(Cys1490Tyr), p.(Glu1087Lys), p.(Thr1526Met), p.(Cys2150Tyr) [7].

### 4.2. Clinical Evaluation

The ophthalmological exam included best-corrected VA (BCVA), which was measured using Snellen charts, a slit-lamp exam, autofluorescence imaging, spectral-domain optical coherence tomography imaging, and ERG testing. Age of onset was defined as the age at which patients first noted decreased VA. One Group 2 patient presented no subjective symptoms at the age of 40 years when retinopathy was already visible in fundus autofluorescence imaging, spectral-domain optical coherence tomography imaging, and ERG testing. Therefore, in this case, the age at the exam when the STGD1 was first noted objectively was considered as the age at onset. Age at the exam was defined as the age of the last ERG recording. Disease duration was calculated as the time between the age at the exam and the age of onset. BCVA was transformed to the logarithm of the minimum angle of resolution (logMAR) equivalent.

### 4.3. Electrophysiology

The ERG was performed with an Espion visual electrophysiology testing system (Diagnosys LLC, Littleton, MA, USA). The ffERG [37] and PERG [36,38] were recorded according to the standards of the ISCEV. The S-cone ERG was recorded according to the ISCEV-approved extended protocol [30]. The recording electrode was an HK loop placed in the fornix of the lower eyelid [39]. A silver chloride reference electrode was placed on the ipsilateral temple, and the ground electrode was positioned on the forehead.

As PERG recording does not require pupil dilation, it was recorded first. It was elicited with a 1° checkerboard pattern with 99% contrast and a temporal frequency of 1.8 Hz, which was presented on a 30.7 × 23.6° (large field) cathode ray tube screen. Peak-to-peak N35 to P50 and P50 to N95 measurements were performed.

Subsequently, the pupils were dilated with 1% tropicamide, and the patients were DA for 20 min, at the end of which ISCEV standard DA 0.01 (flash intensity in cd s m^−2^) and DA 3.0 ffERGs were recorded. Oscillatory potentials were isolated using the off-line digital filtering (100–2000 Hz) of the DA 3.0 wave. Flashes were generated within a full-field ColorDome (Diagnosys LLC, Lowell, MA, USA). LA 3.0 and LA 3.0 flicker ERGs were obtained following 10 min of light adaptation at a background luminance of 30 cd/m^2^.

S-cone ERGs were elicited with 8 cd s/m^2^ blue (449 nm) stimuli on a 140 cd/m^2^ amber (594 nm) background. Due to the very low signal amplitude, at least two series of 80 individual blue-light flashes were averaged to obtain a suitable signal. Standard ERG components were analyzed according to the ISCEV conventions for the measurement of the ERG peak times and amplitudes. Values were considered abnormal if they fell outside our laboratory’s normal range.

The following ERG parameters were selected to perform the statistical analysis: PERG P50 amplitude, representing macular function; DA 0.01 ERG b-wave and DA 3.0 ERG a-wave amplitudes, which mostly represented the rod system function; DA oscillatory potentials; LA 30 Hz flicker ERG and LA 3.0 ERG b-wave, which represented the cone system function; and S-cone ERG.

### 4.4. Statistical Analysis

The data were analyzed using IBM SPSS Statistics software version 27.0 (IBM Corp. Armonk, NY, USA). The Mann–Whitney U Test was applied to compare age during the exam, age at onset, and VA and ERG amplitudes between the two patient groups. Simple linear regression, the correlation between ERG responses, and age at the exam were compared within each patient group. Age was taken as the independent variable, while PERG P50 amplitudes, DA 0.01 ERG b-wave amplitudes, DA 3.0 ERG a-wave amplitudes, oscillatory potentials, LA 30 Hz flicker ERG amplitudes, LA 3.0 ERG b-wave amplitudes, and S-cone ERG amplitudes were taken as dependent variables. A value of *p* < 0.05 was considered to indicate the statistical significance. The patients’ right eyes were used to perform the analysis, as the clinical appearance was very much symmetrical between the eyes.

## 5. Conclusions

In summary, the study quantitatively analyzed individual ERG responses and determined their importance in following disease progression in STGD1 patients depending on genotype. It showed that the most relevant ERG parameters to follow age-related changes in the advanced stage of the disease were DA 0.01 b-wave and DA 3.0 ERG a-wave amplitudes for double-null patients, and the S-cone ERG for patients with other, non-null genotypes. The results are especially important for designing clinical trials that require sensitive and objective biomarkers to determine disease trajectory. The authors propose that S-cone measurements be included in the electrophysiological assessment of patients with STGD1.

## Figures and Tables

**Figure 1 ijms-23-16161-f001:**
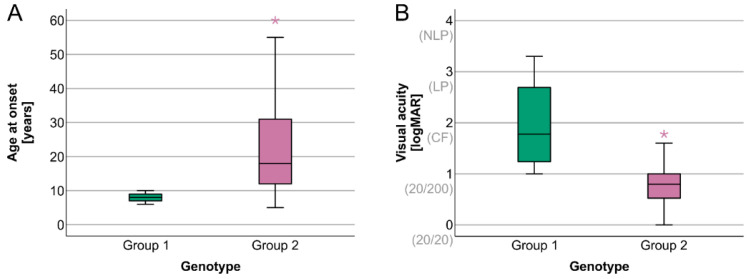
Boxplot chart showing the distribution of the ages at the onset of visual symptoms and visual acuities (VAs) in the two patient groups. Group 1 patients presented a significantly later age of onset (**A**) and significantly better VA (**B**). Horizontal lines represent median values, boxes represent half of the patients, and whiskers represent the remaining data, except in the case of the outliers (stars). LogMAR = logarithm of the minimum angle of resolution.

**Figure 2 ijms-23-16161-f002:**
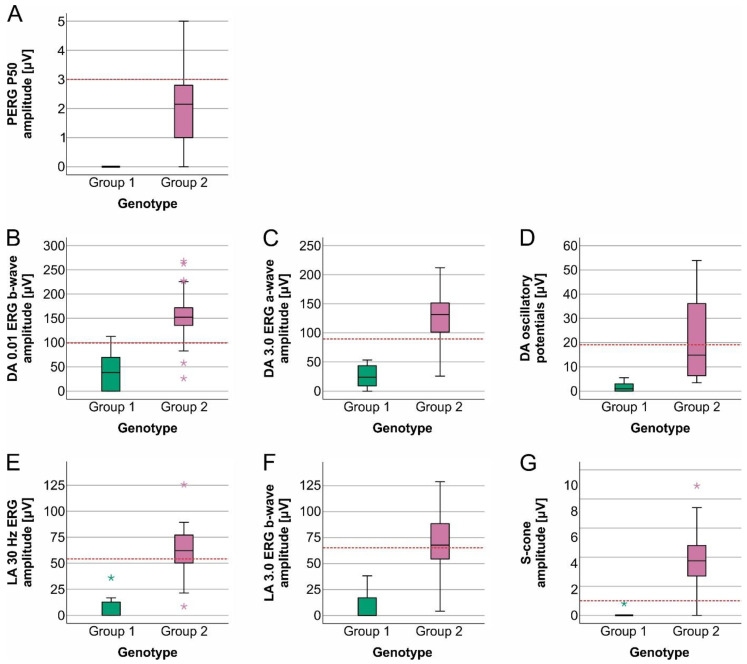
Boxplot charts showing various ERG parameters’ amplitudes in patients with different genotypes. Group 1 patients had significantly lower PERG P50 amplitudes, DA 0.01 ERG b-wave amplitudes, DA 3.0 ERG a-wave amplitudes, oscillatory potentials, LA 30 Hz ERG amplitudes, LA 3.0 ERG b-wave amplitudes, and S-cone ERG amplitudes than Group 2 patients. Note that even though Group 1 patients were of the same age or even younger than Group 2 patients, all their ERG responses were significantly worse. Values were considered abnormal if they fell outside our laboratory normal range (marked with the red, dashed line). Horizontal lines represent median values, boxes represent half of the patients, and whiskers represent the remaining data except in the case of the outliers (stars).

**Figure 3 ijms-23-16161-f003:**
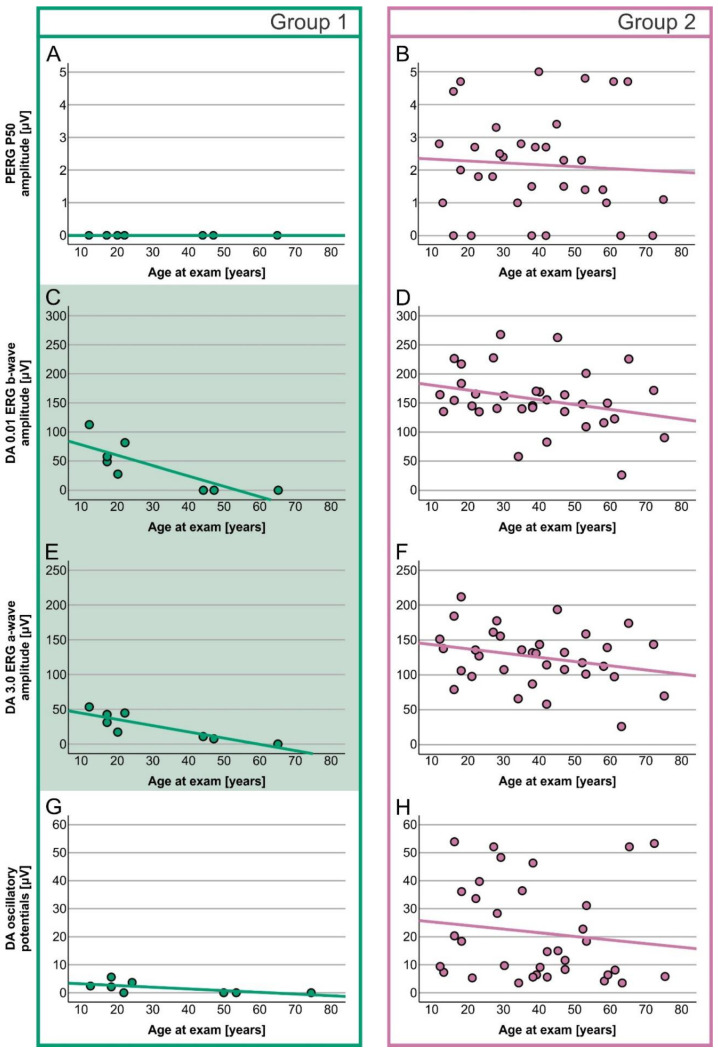
Age-dependent decrease (linear regression) in PERG P50 amplitude and DA ERG responses in Group 1 (**A**,**C**,**E**,**G**) and 2 (**B**,**D**,**F**,**H**) patients. Group 1 is marked with green and Group 2 with purple. Green-colored graphs represent a significant correlation between age and DA 0.01 ERG b-wave and DA 3.0 ERG a-wave amplitudes.

**Figure 4 ijms-23-16161-f004:**
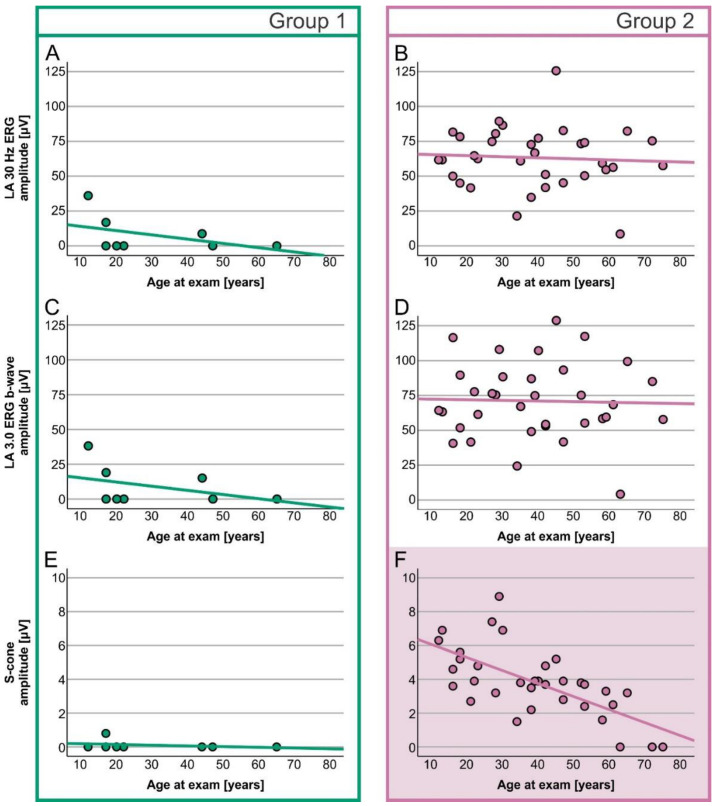
Age-dependent decrease (linear regression) in LA ERG responses in Group 1 (**A**,**C**,**E**) and 2 (**B**,**D**,**F**) patients. Group 1 is marked with green and Group 2 with purple. The purple-colored graph represents a significant correlation between age and S-cone ERG amplitude.

**Figure 5 ijms-23-16161-f005:**
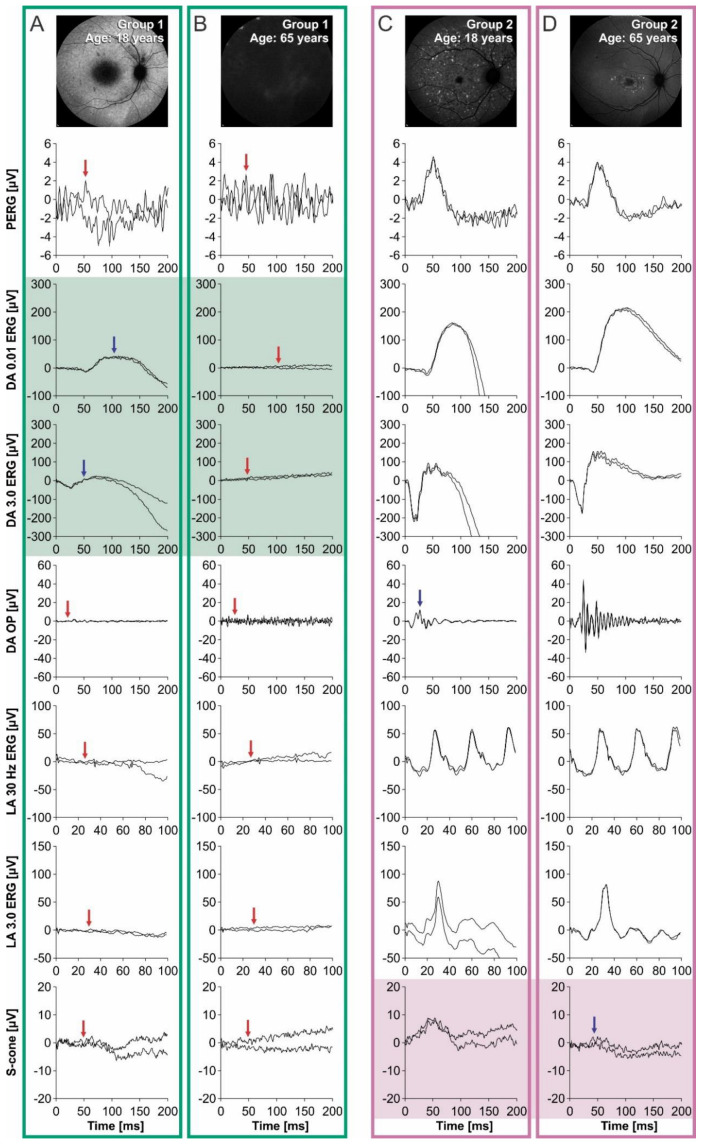
ERG responses from two younger and two older representative patients from each group. Panels (**A**,**C**) show responses from an 18- and 65-year-old patient from Group 1, while panels (**B**,**D**) show responses from an 18- and 65-year-old patient from Group 2. Detectable but abnormal responses are marked with blue arrows, while undetectable responses are marked with red arrows. A fundus autofluorescence image of the macula is shown for each patient in the top-right corner. Note that the DA 0.01 ERG b-wave and DA 3.0 ERG a-wave were only detectable in the 18-year-old patient from Group 1 (**A**) and no responses were detectable in the 65-year-old patient from the same group (**D**). In comparison, all responses were detectable in both the 18- and 65-year-old patients from Group 2 ((**B**) and (**D**), respectively) with, however, a notable difference in the S-cone response.

**Table 1 ijms-23-16161-t001:** Data and results of analyzed clinical characteristics.

Parameter	Group 1 (Median, Range)	Group 2 (Median, Range)	Mann–Whitney Test (U, z, *p* Values)
Age at exam	21 years (12–65 years)	39 years (12–75 years)	U = 174.500, z = 1.234, *p* = 0.222
Age at onset	8 years (6–10 years)	18.5 years (5–60 years)	U = 260.000, z = 3.976, ***p* < 0.001**
Disease duration	12.5 years (4–55 years)	12 years (0–51 years)	U = 104.500, z = −1.010, ***p* = 0.320**
Visual acuity	1.8 (1.0–3.3)	0.8 (0.0–1.8)	U = 28.500, z = −3.463, ***p* < 0.001**
PERG P50 amplitude	0.0 µV (0.0–0.0)	2.2 µV (0.0–5.0)	U = 248.000, z = 3.658, ***p* < 0.001**
S-cone ERG amplitude	0.0 µV (0.0–0.8)	3.8 µV (0.0–8.9)	U = 258.500, z = 3.953, ***p* < 0.001**
DA 0.01 ERG b-wave amplitude	38.3 µV (0.0–122.5)	152.2 (26.5–267.8)	U = 262.000, z = 4.037, ***p* < 0.001**
DA 3.0 ERG a-wave amplitude	24.2 µV (0.0–53.4)	131.4 µV (26.0–211.9)	U = 268.000, z = 4.228, ***p* < 0.001**
Oscillatory potentials	1.1 µV (0.0–5.6)	14.9 µV (3.5–53.9)	U = 265.000, z = 4.135, ***p* < 0.001**
LA 30 Hz ERG amplitude	0.0 µV (0.0–36.0)	62.1 µV (8.5–125.6)	U = 267.000, z = 4.200, ***p* < 0.001**
LA 3.0 ERG b-wave amplitude	0.0 µV (0.0–38.2)	67.8 µV (4.2–128.7)	U = 268.000, z = 4.232, ***p* < 0.001**

ERG = electroretinography; PERG = pattern ERG; DA = dark-adapted; LA = light-adapted. Numerical values are expressed as median (range). U = U test statistics, z = z score. The values in bold indicate statistical significance defined as *p*-value < 0.05.

**Table 2 ijms-23-16161-t002:** Distribution of patients according to the traditional ERG classification.

ERG Group	Patient Group 1 (Two Null Variants)	Patient Group 2 (All Other Genotypes)
Normal ERG		23.5% (8/34)
ERG Group 1 (normal ffERG)	/	47.1% (16/34)
ERG Group 2 (cone dystrophy)	/	17.6% (6/34)
ERG Group 3 (cone-rod dystrophy)	100% (8/8)	11.8% (4/34)

ffERG = full-field electroretinography.

**Table 3 ijms-23-16161-t003:** Results of statistical analysis (simple linear regression) investigating the correlations between age and different ERG parameters within the two patient groups.

ERG Response	Group 1		Group 2	
	Simple Linear Regression		Simple Linear Regression	
	ANOVA Results	B and β Coefficients	ANOVA Results	B and β Coefficients
PERG P50 amplitude	Variable is constant		(F(1, 32) = 0.135, *p* = 0.715, R^2^ = 0.004, R^2^_adjusted_ = −0.027	B = −0.006, 95% CI [−0.037, 0.026]), β = −0.065
DA 0.01 ERG b-wave amplitude	F(1, 6) = 11.134, ***p* = 0.016**, R^2^ = 0.650, R^2^_adjusted_ = 0.591	B = −1.776, 95% CI [−3.079, −0.377]), β = −0.806	F(1, 32) = 2.782, *p* = 0.105, R^2^ = 0.080, R^2^_adjusted_ = 0.041	B = −0.606, 95% CI [−1.402, 0.189]), β = −0.283
DA 3.0 ERG a-wave amplitude	F(1, 6) = 17.839, ***p* = 0.006**, R^2^ = 0.748, R^2^_adjusted_ = 0.706	B = −0.897, 95% CI [−1.417, −0.474]), β = −0.865	F(1, 32) = 2.410, *p* = 0.130, R^2^ = 0.070, R^2^_adjusted_ = 0.051	B = −0.833, 95% CI [−1.849, 0.184]), β = −0.265
Oscillatory potentials	F(1, 6) = 4.103, *p* = 0.089, R^2^ = 0.406, R^2^_adjusted_ = 0.307	B = −0.071, 95% CI [−0.157, 0.015]), β = −0.637	F(1, 32) = 0.558, *p* = 0.461, R^2^ = 0.017, R^2^_adjusted_ = −0.014	B = −0.128, 95% CI [−0.479, −0.222]), β = −0.131
LA 30 Hz ERG amplitude	F(1, 6) = 1.508, *p* = 0.265, R^2^ = 0.201, R^2^_adjusted_ = 0.068	B = −0.306, 95% CI [−0.916, 0.304]), β = −0.448	F(1, 32) = 0.124, *p* = 0.727, R^2^ = 0.004, R^2^_adjusted_ = −0.027	B = −0.076, 95% CI [−0.514, 0.326]), β = −0.062
LA 3.0 ERG b-wave amplitude	F(1, 6) = 1.170, *p* = 0.321, R^2^ = 0.163, R^2^_adjusted_ = 0.024	B = −0.300, 95% CI [−0.978, −0.378]), β = −0.404	F(1, 32) = 0.537, *p* = 0.868, R^2^ = 0.001, R^2^_adjusted_ = −0.030	B = −0.046, 95% CI [−0.599, 0.508]), β = −0.030
S-cone ERG amplitude	F(1, 6) = 0.537, *p* = 0.491, R^2^ = 0.082, R^2^_adjusted_ = −0.071	B = −0.004, 95% CI [−0.018, −0.010]), β = −0.287	F(1, 32) = 25.810, *p* < 0.001, R^2^ = 0.446, R^2^_adjusted_ = 0.429	B = −0.077, 95% CI [−0.108, −0.046]), β = −0.065

β = standardized regression coefficient, B = unstandardized regression coefficient, brackets denote 95% confidence intervals. The values in bold indicate statistical significance (marked with green for Group 1 and purple for Group 2).

## Data Availability

The data that support the results of this study are available upon request from the corresponding author, A.F. The data are not publicly available due to personal data protection policies.

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
