# Peer review of "Electroretinography as a Biomarker to Monitor the Progression of Stargardt Disease"

_ijms, 2022, doi:10.3390/ijms232416161_

Round 1
Reviewer 1 Report
The article is well done and the research is relevant.
Just a few comments to the author.
The authors divide the groups according to the genotype. Group 1 consisted in patients with null variants, and They considered to be null if it was either a stop variant a frame-shifting variant resulting in a stop codon, or splitting and missense variants previously shown to behave as null.
Simonelli uses the age of onset of the disease to classify in severe with an early onset (before the 20 years old) and mild if the age of onset were later. And they stablish and association with the ERG.
Simonelli found missense variants in both groups.
We ask the authors if they considered to divide the groups by the phenotype and the onset of the disease and if the results could be similar.
You have found any association between the fundus appearance and the severity of the disease.
Minor change
In Figure 3. For group 2 patients, dependency between age and S-cone amplitude is shown in c (and no in B)
Reviewer 2 Report
Electroretinography as a biomarker for monitoring Stargardt disease
This article studies the potential of electroretinogram (ERG) as a quantifiable marker for the progressive degeneration in Stargardt disease. They studied 42 patients which were classified according to genotype: 8 patients with double null ABCA4 variants (Group 1) and 34 patients with other ABCA4 genotypes (Group 2).
As well as age at exam, age at onset and visual acuity, the authors performed 7 different types of erg on the patients, and found that they could find 3 different biomarkers of disease progression, a substantial improvement on the standard classification, which does not present an age-prognosis correlation.
These consisted of 2 dark-adapted ERG measures (rod-system) for the progression of the disease in Group 1 patients, and S-cone ERG for progression of the disease in Group 2 patients.
This study has the potential to advance the field of Stargardt disease diagnosis, but can be substantially improved with a bit more data analysis in the data presented.
Comments
- I liked the use of color to differentiate between the 2 groups in the Figures. I think that Figure 3 should be bigger (7 plots instead of 3), with the results of each ERG test as individual values, and both groups on each plot, separated by color. This would help the reader appreciate the distribution in samples, as well as the enhance the linear regression observed in the 3 plots currently displayed in Figure 3.
- The authors gathered a very rich dataset consisting of 7 ERG measures, disease onset and visual acuity. It would substantially increase the power of their observations if they could reduce the dimensionality of their data, possibly through principal component analyses (PCA). I suggest adding at least 1 PCA plot to the study, in the hopes that further classification of disease progression can be found.
Minor Comments
- I suggest including the word “progression” in the title, to help inform the prospective reader.
- I suggest adding a sentence explaining briefly the function of the ABCA4 protein in the intro
- In the intro or discussion, say how long it takes to take a measure of ERG, and compare it to the 7 measures taken in the study and also compare to fundus or slit-lamp imaging.
- Figure 3 legend refers to panel B instead of C (line 121)
Line 115, add a comma after “Group 2 patients”.
